



**Impact of meteorological conditions on BVOC emission rate from Eastern**
**Mediterranean vegetation under drought**
Qian Li[1], Gil Lerner[1], Einat Bar[2], Efraim Lewinsohn[2], Eran Tas[1*]
[1] Institute of Environmental Sciences, The Robert H. Smith Faculty of Agriculture, Food and
Environment, The Hebrew University of Jerusalem, P.O. Box 12, Rehovot 7610001, Israel
[2] Department of Vegetable Research, Agricultural Research Organization – Newe Ya'ar Center,
Israel
* Correspondence to:
Eran Tas, Institute of Environmental Sciences, The Robert H. Smith Faculty of Agriculture, Food
and Environment, The Hebrew University of Jerusalem, P.O. Box 12, Rehovot 7610001, Israel
eran.tas@mail.huji.ac.il





**Abstract**
A comprehensive characterization of drought's impact on biogenic volatile organic
compounds (BVOC) emissions is essential for understanding atmospheric chemistry under
global climate change, with implications for both air quality and climate model simulation.
Currently, the effects of drought on BVOC emissions are not well characterized. Our study
aims to test: i) whether instantaneous changes in meteorological conditions can serve as a
better proxy for drought-related changes in BVOC emission compared to the absolute
values of the meteorological parameters, as indicated in a companion article based on
BVOC mixing-ratio measurements; ii) the impact of a plant under drought stress receiving
a small amount of precipitation on BVOC emission rate, and on the manner in which the
emission rate is influenced by meteorological parameters. To address these objectives, we
conducted our study during the warm and dry summer conditions of the Eastern
Mediterranean region, focusing on the impact of drought on BVOC emissions from natural
vegetation. Specifically, we conducted branch-enclosure sampling measurements in Ramat
Hanadiv Nature Park, both under natural drought and after irrigation (equivalent to 5.5–7
mm precipitation), for six selected branches of *Phillyrea latifolia*, the highest BVOC
emitter in this park, in September–October 2020. The samplings were followed by gas
chromatography-mass spectrometry analysis for BVOCs identification and flux
quantification. The results corroborate the finding that instantaneous changes in
meteorological parameters, particularly relative humidity (RH), offer the most accurate
proxy for BVOC emission rates under drought, compared to the absolute values of either
temperature (T) or RH. However, after irrigation, the correlation of the detected BVOC
emission rate with the instantaneous changes in RH became significantly more moderate,



or even reversed. Our findings highlight that under drought, the instantaneous changes in
RH, and to a lesser extent in T, are the best proxy for the emission rate of monoterpenes
(MTs) and sesquiterpenes (SQTs), whereas under moderate drought conditions, T or RH
serves as the best proxy for MT and SQT emission rate, respectively. In addition, the
detected emission rates of MTs and SQTs increased by 150% and 545%, respectively, after
a small amount of irrigation.

**1 Introduction**
Biogenic volatile organic compounds (BVOCs) are released by plants and other organisms
to the atmosphere. They play a critical role in both climate change and photochemical air
pollution (Cai et al., 2021; Calfapietra et al., 2013; Curci et al., 2009; Guenther, 2013;
Kesselmeier and Staudt, 1999; Peñuelas et al., 2009). BVOCs are thought to be emitted by
plants as a defense mechanism against biotic and abiotic stresses, such as herbivory and
high temperatures (Berg et al., 2013; Blande et al., 2007; Brilli et al., 2009; Peñuelas and
Munné-Bosch, 2005). BVOCs may also be involved in plant–plant and plant–animal
communication, allowing plants to signal to other organisms about their response to
environmental conditions (Baldwin et al., 2006; Filella et al., 2013; Niinemets and Monson,

2013).

The emission rate and composition of BVOCs can vary widely depending on

various factors, such as meteorological conditions, rate of synthesis, and physicochemical
properties (Niinemets and Monson, 2013). Climate change is expected to significantly
impact BVOC emission rate and composition. As temperature rises, the emission rate of
most BVOCs increases in an Arrhenius-type manner (Goldstein et al., 2004; Greenberg et



al., 2012; Guenther et al., 1995; Monson et al., 1992; Niinemets et al., 2004; Tingey et al.,
1990). On the other hand, drought can have a more complex effect on the emission and
composition of BVOCs. Depending on the type of vegetation, the level of drought stress,
and additional ambient conditions, the emission of BVOCs can be partially or completely
suppressed (Fortunati et al., 2008; Holopainen and Gershenzon, 2010; Llusia et al., 2016;
Peñuelas and Staudt, 2010; Schade et al., 1999), or enhanced in a way that has not yet been
characterized (Fitzky et al., 2023; Geron et al., 2016; Potosnak et al., 2014).
The effect of drought on isoprene emission has been extensively studied, and it was
discovered to be postponed relative to, and/or less significant than the effect on
photosynthetic rate (Asensio et al., 2007; Brilli et al., 2007; Fortunati et al., 2008;
PEGORARO et al., 2006; Ryan et al., 2014). However, whereas under moderate drought
stress, isoprene emission may only slightly decrease or increase, it was shown to decrease
considerably under severe or prolonged drought stress (Fortunati et al., 2008; Han et al.,
2022; Jiang et al., 2018). The impact of drought on the emission of other BVOCs, such as
monoterpenes (MTs) and sesquiterpenes (SQTs), has been less studied.
The Eastern Mediterranean has a unique climate characterized by a hot and dry
summer, making it an ideal location to study the impact of drought on BVOC emissions.
The semiarid and arid regions are particularly vulnerable to climate change, and climate
simulations predict that the Eastern Mediterranean region will experience more frequent
and severe droughts in the future (Giorgi and Lionello, 2008; Lionello, 2012). Research
conducted in Israel has investigated the impact of drought on BVOC emissions from a
range of local plant species. For example, Llusia et al. (2016) examined the effect of
drought on terpene emission from Yatir Forest, a pine forest in the northern Negev. They



found that some of the MT and SQT emissions increased under moderate drought
conditions but strongly decreased under severe drought conditions. Another measurement
by Li et al. (2023), performed in late autumn 2016 in Shibli Forest in northern Israel, found
that under severe drought stress, BVOC emissions respond more significantly to the
instantaneous changes in meteorological parameters (especially relative humidity [RH])
than to the meteorological parameters themselves. These studies suggest that the impact of
drought on BVOC emissions is not well-characterized and varies in a complex manner,
depending on plant species, BVOC type, and meteorological parameters, such as
temperature (T) and RH, as well as the level of drought stress. Hence, more research is
needed to better characterize the effect of drought on BVOC emission rates and
composition, which can in turn improve air quality and climate modeling.

In this study, we use the severe drought conditions during the autumn in the Eastern

Mediterranean to study the effect of drought on the emission of BVOCs from natural
vegetation. The main specific objectives of this study were to: i) identify whether
instantaneous changes in meteorological parameters can serve as a better proxy for BVOC
emission rates under drought than their absolute values, and ii) determine the extent to
which small precipitation amounts, under drought conditions, can impact BVOC emission
rates and the manner in which the emission rate is influenced by meteorological parameters.

**2 Methods**
We used an enclosure-based measurement system to quantify BVOC emissions, allowing
for direct measurement of BVOC fluxes at the branch level. The measurements were
performed in autumn under the prolonged drought stress conditions typical to this region.



BVOC measurements in the Eastern Mediterranean are rare, and to the best of our
knowledge, our study is the first to apply direct measurements of BVOC flux from specific
branches of natural vegetation in this region. Plants were sampled before and after the
application of a small amount of irrigation to study the response of BVOC emissions, under
exposure to natural drought conditions, to a small amount of precipitation. This was
followed by gas chromatography–mass spectrometry (GC–MS) to identify and quantify
the emitted BVOCs. Closed chambers are often used for measurements of BVOCs at the
branch level (Duhl et al., 2008). Compared to open-system methods, the enclosure-based
system (including a glass cuvette or Tedlar bag) can focus on specific vegetation in a more
controlled manner. To investigate the effects of drought on BVOC emission rates and
composition, we performed two sets of measurements – before and after irrigation – for
comparison. To study the effect of meteorological conditions on BVOC flux, we monitored
meteorological parameters inside the bag and at a meteorological station that was 300–600
m from the branches.

***2.1 Sampling site and studied species***
The on-site branch measurements were conducted at Ramat Hanadiv Nature Park (32°
33′ 19.87″ N, 34° 56′ 50.23″ E), 3.6 km from the Eastern Mediterranean seashore and
exposed to a typical Eastern Mediterranean climate, with annual precipitation of 640 mm
(averaged over the last 5 years, and occurring mainly between November and March). The
vegetation at the site is dominated by mixed Mediterranean shrubbery. More details about
the site and vegetation can be found in Li et al. (2018) and Dayan et al. (2020). The
measurements were conducted at the end of summer/beginning of autumn under drought



conditions. No precipitation was recorded for 108 days between 24 May 2020 and the
beginning of the study on 9 Sep 2020.

*Phillyrea latifolia* (broad-leaved phillyrea), identified as the greatest BVOC-

contributing plant species in the Ramat Hanadiv natural park, was sampled. The species is
native to the Mediterranean Basin and belongs to the family Oleaceae. In Ramat Hanadiv,
it accounts for 7.5% of all vegetation, but up to ~35% of all BVOC emissions, according
to the Model of Emissions of Gases and Aerosols from Nature (MEGAN v2.1; Dayan et
al., 2020; Guenther et al., 2012; Li et al., 2018). The selected plants were mature and did
not show any visible signs of senescence. Sampled branches were shaded, to eliminate the
effect of non-natural high temperature in the enclosure system, and measurements were
performed at 1.5 to 2 m aboveground.

***2.2 Branch-enclosure sampling system and setup***
Figure 1 presents a self-made branch-sampling system was used for this study. All tubes
and connections are Teflon, while valves and flowmeters are stainless steel. A compressor
provides a controllable rate of ambient air flow through an adjustable T-junction valve (to
adjust the flow rate) to a zero-air device (Model 1150 dual reactor, Thermo Fisher
Scientific, Waltham, MA, USA), which includes a catalytic converter heated to ~350 °C to
oxidize CO and HC to $CO_2$ and $H_2O$. From the zero-air device, the air flows through a
copper coil to cool it down, and then through a mass flowmeter into a Tedlar bag (CEL
Scientific Corporation, Cerritos, CA, USA), at a flow rate of about 7 L min$^{-1}$ (monitored
by flowmeter A), a high enough inflow to produce slight overpressure inside the bag. The
inert and light-transparent 10 L Tedlar bag is tied tightly around a tree branch, along with



an EL-MOTE-TH temperature and RH sensor (Lascar Electronics, Whiteparish, Wiltshire,
UK). The outlet airflow (~4 L min⁻¹ monitored by flowmeter B) is directed to the C2-
CAXX-5032 hydrophobic inert-coated stainless-steel adsorbent tube (CSLR, Markes
International, Llantrisant, UK) precoated with a mixture of Tenax TA and Carbograph as
adsorbent, at a rate of ~0.2 L min⁻¹ (monitored by flowmeter C), regulated by the T-junction
valve downstream of flowmeter B. The flow rate through the adsorbent tube, as well as T
and RH were recorded with a CR1000 data logger (Campbell Scientific, Logan, UT, USA).

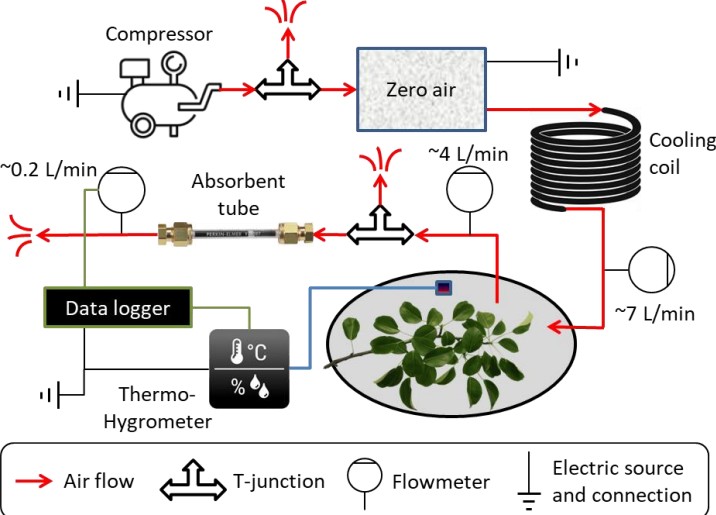

**Figure 1**. Schematic of the branch-enclosure sampling system. VOCs are removed from the ambient air
before entering a transparent Tedlar bag and an adsorbent tube to monitor BVOC emissions from the enclosed
branch, using a flow-controlled system (see Sect. 2.2).

***2.3 Analytical quantification of the sampled BVOCs***
A Centri™ (Markes International) preconcentration system was used to desorb the tubes
into the cold trap (graphitized carbon trap; used for sampling VOCs of C4/5 to C30/32)
under the following conditions: desorption for 5 min at 280 °C with a trap flow of 30 mL
min⁻¹. Desorption of trap was at a rate of 20 °C s⁻¹ to 300 °C into an Agilent GC–MS



(7890A/5975C) system (Santa Clara, CA, USA) equipped with a Stabilwax column
(Restek, 30 m, 0.25 mm ID capillary column; polyethylene glycol, 0.25 µm film thickness).
The general run parameters were as follows: injector, 230 °C; column oven, initial
temperature of 45 °C for 5 min, followed by a ramp of 5 °C min$^{-1}$ to 120 °C, 20 °C min$^{-1}$
to 240 °C final, and 5-min hold with a total run time of 31.5 min; carrier gas, He 32 psi;
mass spectrometer ionization energy, 70 eV; m/z, 41 to 300; scan time, 5.4 s. The
chromatograms were analyzed using MassHunter Quant Analysis (B.10.00, Agilent
Technologies, Santa Clara, CA, USA) software. Compounds were identified by comparing
their relative retention indices and mass spectra with those of authentic standards or those
found in the literature, supplemented with W10N14 and 2205 GC–MS libraries.

We chose to analyze the most abundant BVOC species: *cis*-β-ocimene (E, Z) (MT),

and β-caryophyllene, α-humulene, α-farnesene, germacrene-D (SQTs). For calibration,
analytical-grade standard solutions (7–12 concentrations) were prepared, ranging in
concentrations from 0.25 to 1000 ng mL$^{-1}$ by diluting known masses of pure chemicals
with methanol. The calibration analytes were injected using a GC syringe onto clean
sorbent tubes connected to a calibration solution-loading rig (Markes International) at a
nitrogen flow of 80 mL min$^{-1}$. The standards for the BVOC species were *cis*-β-ocimene (E,
Z) (W353977, Sigma-Aldrich) (MT), and β-caryophyllene (22075-1ML-F, Sigma-
Aldrich), α-humulene (PHL83351, Sigma-Aldrich), α-farnesene (Biosynth® Carbosynth
Ltd., UK), germacrene-D (Toronto Research Chemicals, Canada) (SQTs). All standard-
loaded tubes were prepared in triplicate and results were averaged. The loaded tubes were
analyzed under the same conditions used for the other samples. Standard curves of peak
area counts vs. VOC mass (µg) were fitted using linear regression analyses; both yielded





high regression coefficients ($r^2 \geq 0.99$ in most cases). More details on the calibration are
provided in Sect. S1.

*2.4 Experimental setup*
*2.4.1 Branch sampling, meteorological parameter measurements and flux evaluation*
The field measurements were performed from late summer to early autumn – 9 Sep to 27
Oct 2020. Samplings were conducted on six selected *Phillyrea latifolia* branches on
different bushes. Each branch was measured over two sequential days: 8–9 Sep, 14–15 Sep,
22–23 Sep, 12–13 Oct, 19–20 Oct, and 26–27 Oct. The bushes were at least 20 m apart, to
enable selective irrigation for individual shrubs. Meteorological parameters were measured
at a distance of 300–600 m from the branch measurements. These parameters included T
and RH, measured using a Campbell HC2S3 probe; net radiation, measured with a CNR4
Kipp & Zonen net radiometer; and wind speed and direction, recorded by a 05103 R.M.
Young sensor. Eight 30-min samplings were performed per measurement day. In addition,
two reference samplings were performed with full equipment setup, but no branch inside
the bag. These reference samplings were performed before and after the eight
measurements. Prior to the first reference sampling, the system and branches were given
at least 60 min to adapt to the different conditions after the setup of the bag and equipment.
Following the 10th sampling on the second measurement day of each 2-sequential-day
period, the sampled branch was cut and sent to the laboratory for leaf analysis. Leaf net dry
weight and area were evaluated within 24 h after cutting the branch. All leaves were
scanned, and a digital color-based image-processing method was used to identify the total
(RGB values: 40–200, 50–200, 30–200) and healthy (RGB values: 40–110, 50–105, 30–



80) leaf areas. The leaves were then dried for 72 h at 60 °C, and their dry weight was
recorded.
The sampling tubes were kept in a cooler with a temperature below 5 °C after the
measurement, and analyzed within 5 days of sampling by GC–MS (see Sect. 2.3). Of the
identified species, the MT and four SQT compounds with the highest sampled mass (*cis-*
β-ocimene, β-caryophyllene, α-humulene, α-farnesene, and germacrene D) were chosen
for quantification by GC–MS (see Sect. 2.3).
The emission rate of BVOCs per leaf area, $E_A$ (ng cm$^{-2}$ h$^{-1}$), for a branch was
evaluated by the following formula:

$$E_A = \left( m \frac{F_{in-B}}{F_{out-T}} \right) / (A \cdot t) \tag{1}$$

where $m$ (ng) is the evaluated mass of any BVOC compound inside the tube, $F_{in-B}$ (L min$^{-1}$
) and $F_{out-T}$ (mL min$^{-1}$) are the flow rate pumped into the bag and the flow rate through
the adsorbent tube, respectively, A (cm$^2$) is the total leaf area of the branch, and $t$ (h) is the
sampling time.
The emission rate of BVOCs per biomass, $E_M$ (ng g$^{-2}$ h$^{-1}$), was evaluated by:

$$E_M = \left( m \frac{F_{in-B}}{F_{out-T}} \right) / (M \cdot t) \tag{2}$$

where M (g) is the leaf biomass of the branch.

***2.4.2 Irrigation and soil-water content quantification***
Manual irrigation was applied at the end of the first measurement day of each 2-sequential-
day measurement period (see Fig. 2). The irrigation amounts were 50–70 L within a radius
of 1–2 m from the stem of the plants used for sampling (equivalent to 5.5–7 mm rain). This



irrigation served to identify the potential effect of a small precipitation event during a
drought period on BVOC emission rate and composition.
Ten soil samples were collected at solar noon time within 2 m from the sampled
plant on every experimental day. To evaluate the soil-water content, soil samples were
weighed on the day of collection, and weighed again after drying them in an oven at 105 °C
for 24 h. The following formula was used to calculate the soil-water content:
$$w = \frac{M_{tot} - M_{dry}}{M_{dry}} \times 100\% \qquad (3)$$
where w (g/g) is the soil gravimetric water content and $M_{tot}$ (g) and $M_{dry}$ (g) are the total
and dried soil mass, respectively.

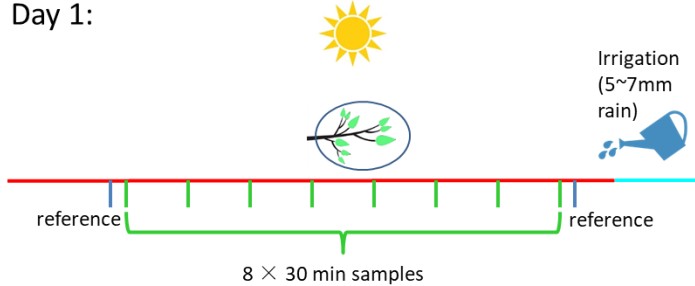

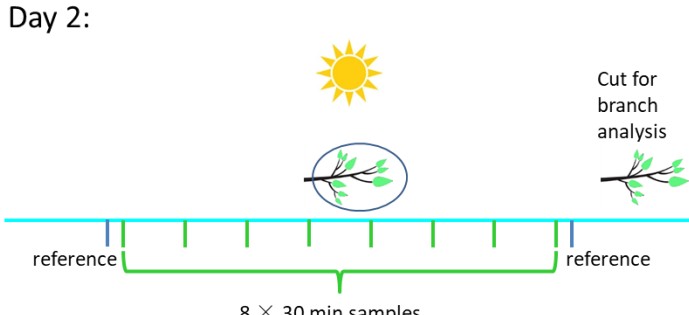

**Figure 2.** Schematic of the experimental design. Day 1 and Day 2 represent, respectively, the first and
second day of each two-sequential-day sampling period for a specific branch. Green and blue bars represent
sampling measurements and reference measurements, respectively. The red and cyan lines mark sampling
prior to manual irrigation on Day 1 and after manual irrigation, on Day 2, respectively.



### 2.4.3 Correlation between BVOC emission rate and temporal changes in RH and T


To test the effect of instantaneous changes in RH and T on the emission rate of the sampled
BVOCs, we studied the correlation between the temporal changes in both ambient air RH
and T with the BVOC emission rate during the sampling. BVOC sampling length was 30
min, with a gap of 1 h between each sampling. To account for instantaneous changes in
RH and T we introduce $\delta_{RH}$ and $\delta_T$, respectively. $\delta_{RH}$ is defined as follows:
$$\delta_{RH} = \sum_{i=1}^{n} \left( \frac{RH_{i+1}}{RH_i} - 1 \right) \tag{4}$$

where $i$ is the 10 min time step, and $n$ is the number of time steps.
$\delta_T$ is defined in the same manner as follows:
$$\delta_T = \sum_{i=1}^{n} \left( \frac{T_{i+1}}{T_i} - 1 \right) \tag{5}$$

The correlations between $\delta_{RH}$, $\delta_T$ and the BVOC fluxes for all samples were tested
for different values of $n$. In a preliminary test, it was found that the highest average
correlations of $\delta_{RH}$ and $\delta_T$ with BVOC emission rate were obtained when $n = 9$.
Accordingly, the calculation duration of $\delta_{RH}$ and $\delta_T$ began 60 min before each 30 min
BVOC emission rate sampling. This finding is consistent with a similar analysis conducted
by Li et al. (2023). Similarly, the correlation between $\delta_{RH}$ and $\delta_T$ and BVOC emission rate
in that study applied $\delta_{RH}$ and $\delta_T$ which were calculated for 90 min cycles, while the
beginning of each cycle was 60 min prior to the beginning of each compatible 30 min
BVOC sampling.

### 2.4.4 Afternoon emission trend (AET) analysis


Under drought conditions, the increased stomatal resistance can largely reduce the BVOC
emission rate (see Sect. 1). Accordingly, it was found that the BVOC mixing ratio tends to





reach a minimum around noontime when RH tends to reach its daily minimum and stomatal
conductance is limited (Nobel, 1999), and then gradually increase in the afternoon (Li et
al., 2023). Our observations indicated a clear increase in BVOC emission rates during the
afternoon for the days before the irrigation. On those days, no clear decrease in BVOC
emission was observed before noon; instead, the BVOCs generally exhibited lower
emission rates.  Here we introduce a method for quantifying the trend of emission rate right
after the mid-day minimum, which applies the afternoon emission trend (AET) index:
$$\text{AET} = \sum_{i=1}^{n} \left( \frac{E_{i+1}}{E_i} - 1 \right) \tag{6}$$

where $E_i$ is the emission rate of the $i_{\text{th}}$ sample, while $i = 1$ indicates the daily minimum
around noontime, between 12:00–14:00 h. Hence, the AET indicates the trend and
magnitude of the emission in the afternoon of any measurement day.

**3 Results and discussion**
*3.1 Analysis of branch leaves*
Figure 3 shows the total leaf area (cm$^2$), green leaf area (cm$^2$), leaf water content, and soil
moisture before and after irrigation of each sampling branch. Leaf green area ranged
between 68% to 89% of the total leaf area. Soil moisture was around 12.5–14.0% before
irrigation and ~14.3–26.2% after irrigation. Interestingly, the leaf water content after
irrigation increased gradually during the experimental period, indicating that the capacity
for water uptake from the soil increases with drought prolongation.

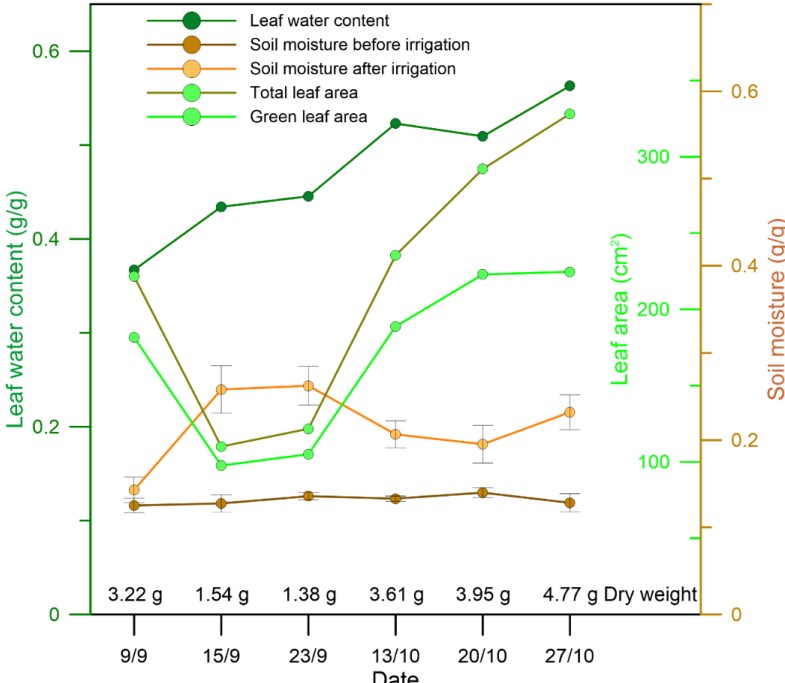

**Figure 3.** Properties of the sampled branch leaves and soil moisture within a radius of 1 m from the stem of the sampling plant. Presented leaf property values are averages over all sampled branch leaves.

### *3.2 Emission rates of MTs and SQTs*

Whereas previous branch enclosure studies focused primarily on isoprene emissions (Genard-Zielinski et al., 2015; Genard-Zielinski et al., 2018; Saunier et al., 2017), our measurements did not detect large amounts of isoprene emissions from the selected *Phillyrea latifolia*, in line with previous studies showing that some plant types do not emit notable amounts of isoprene (Aydin et al., 2014; Bracho-Nunez et al., 2013). Our analysis focused on the MTs and SQTs detected in our observations, as described in the following section.





### 3.2.1 MTs

On all 10 sampling days for which MTs were identified, the 5 days prior to irrigation were under drought conditions (i.e., more than 100 days after the last precipitation event), and 5 days were under irrigation conditions on the same branches (see Sect. 2.4.2). The branch which was sampled on Sep 14–15 did not show any detectable MT emission. The diurnal emission fluxes of MTs from the branches are shown in Fig. 4.

The daily average emission rate of MTs over all sampling days ranged from 11.7–2151.4 ng cm$^{-2}$ h$^{-1}$ (0.89–121.5 µg g$^{-1}$ h$^{-1}$), with *cis*-β-ocimene being most abundant at 88% of all detected MTs. These MT emission rates are similar to previous branch enclosure studies, which were conducted predominantly between May and October under Western Mediterranean conditions, where they ranged from 0 to approximately 140 µg g$^{-1}$ h$^{-1}$ (Bracho-Nunez et al., 2013; Llusià and Peñuelas, 2000; Núñez et al., 2002; Owen et al., 1997; Owen and Hewitt, 2000; Staudt et al., 2001; Street et al., 1997). Less information is available on the emission rates of MTs in the Eastern Mediterranean. Aydin et al. (2014) used a branch enclosure system to detect emission rates ranging from 0.0047 to 14.2 µg g$^{-1}$ h$^{-1}$ in 14 different forested areas in Turkey. Seco et al. (2017) quantified MT emissions using eddy covariance method in pine forests in Israel, studying a semiarid site (Yatir) and a Mediterranean sub-humid site (Birya) in the spring. Emission fluxes were found to average at 40 ng cm$^{-2}$ h$^{-1}$ (Yatir) and 100 ng cm$^{-2}$ h$^{-1}$ (Birya), with peak values of 100 (Yatir) and 190 (Birya) ng cm$^{-2}$ h$^{-1}$, while the daytime standardized MT emission capacities were similar across both sites.

In our study, MT emissions under drought conditions ranged from 11.7 ng cm$^{-2}$ h$^{-1}$ to 499.0 ng cm$^{-2}$ h$^{-1}$, which is somewhat higher than other values reported in the Eastern

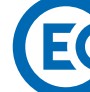

Mediterranean. It is important to note that differences in emission rates between our study
and the previously reported values in this region might be attributed to the different
measurement methodologies employed. Following irrigation, the mean daily MT emission
rates increased in four out of the five investigated branches, and ranged from 13.6 ng cm$^{-2}$
h$^{-1}$ to 2151.4 ng cm$^{-2}$ h$^{-1}$. This reflects an average 150% increase for all sampling days in
the range of emission rates following irrigation, indicating that even a small amount of
water during a period of drought stress can significantly influence MT emissions. This
effect may be related to the dramatic increase in stomatal conductance, due to the increase
in water availability following irrigation  (Medrano et al., 2002; Miyashita et al., 2005;
Vilagrosa et al., 2003).
AET (Sect. 2.4.4) values specified in figures 4 and 5 reinforced the significant
effect of small irrigation amounts on BVOC emission rates under drought, considering that
on drought days, AETs were high and positive, whereas after irrigation, AETs became
moderate or negative. This observation is consistent with previous studies showing that the
emission of BVOCs can be affected by the vegetation's stomatal activity, which tends to
be lower around noontime during drought stress (Li et al., 2023; Seco et al., 2017). Stomatal
resistance is typically two orders of magnitude larger than cuticular resistance (Nobel, 1999)
and therefore, the midday minimum and the following increase in MT emissions under
drought conditions may be mostly due to stomatal resistance, which can limit the exchange
of gases between the plant and the atmosphere. In other words, the increased emission of
MTs after irrigation may be due to reopening of the stomata, which allows for the release
of VOCs.




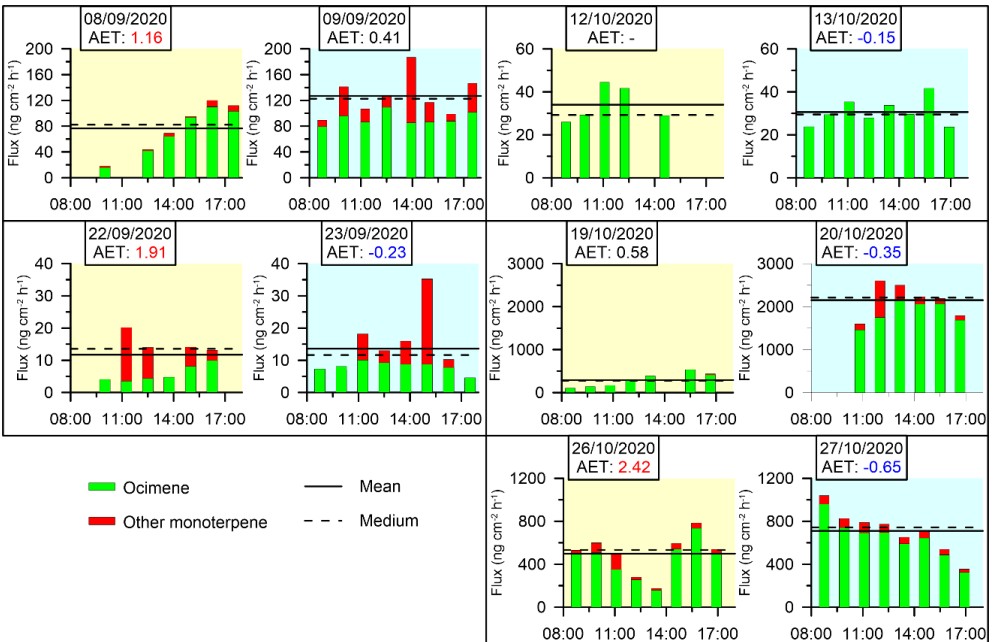

**Figure. 4** Branches' diurnal MT emission fluxes. No MTs were detected for the branch sampled on 14–15

Sep. Yellow and blue shading indicate the days before and after irrigation, respectively (see Sect. 2.4.2).

Horizontal solid and dashed lines are daytime mean and median fluxes of MTs, respectively. AET values

(see Sect. 2.4.4) are marked in red and blue when they are larger than 1 or negative, respectively.

### *3.2.2 SQTs*

Figure 5 shows the emission fluxes of SQTs for the branches under drought and irrigation

conditions. The four major SQTs detected were β-caryophyllene, α-humulene, germacrene

D, and α-farnesene. The daily average emission rate of SQTs ranged from 1.7–2595.7 ng

$cm^{-2}$ $h^{-1}$ (0.11–146.6 µg $g^{-1}$ $h^{-1}$). In contrast to MTs, few studies provide branch enclosure

measurements for SQTs. Notably, our study found significantly higher emission rates than

previous research conducted between June and October under Eastern Mediterranean

conditions, where rates ranged from 0.0011 to 0.63 µg $g^{-1}$ $h^{-1}$ (Aydin et al., 2014; Bracho-

Nunez et al., 2013). The emission fluxes of the SQTs were overall comparable to those of



the MTs, which is a notable finding, considering that SQT emission rates are frequently
around a quarter of the MT flux (Saunders et al., 2003; Sindelarova et al., 2014). The
finding of relatively high SQT emission rates appears to be in line with the findings of Li
et al. (2023), who reported relatively high mixing ratios of SQTs (33.6 times higher than
isoprene, and 18.9 times higher than MTs) under drought conditions in the same region.

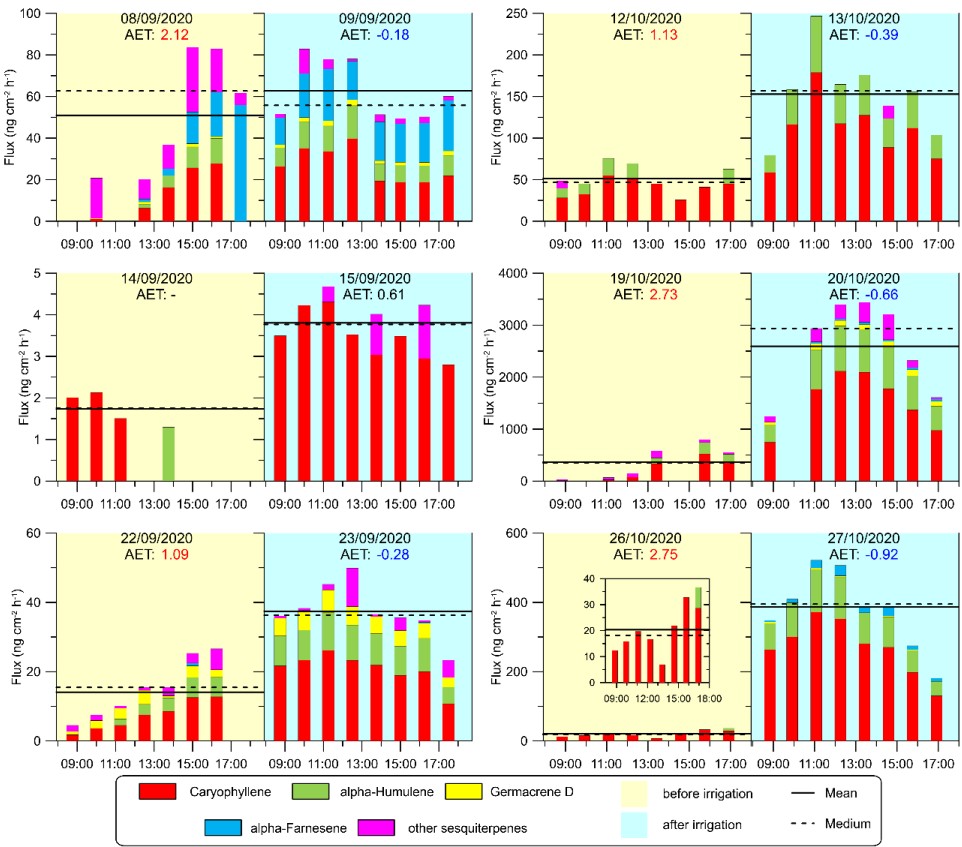

**Figure. 5** Diurnal SQT emission fluxes from the sampled branches. Column colors represent the emission
fluxes of four types of SQTs, and the magenta section of the columns refers to other SQTs. Yellow and blue
shading indicate the days before and after irrigation, respectively (see Sect. 2.4.2). Horizontal solid and
dashed lines are daytime mean and median SQT flux rates, respectively. AET values (see Sect. 2.4.4) are
marked in red and blue when they are larger than 1 or negative, respectively. To better present the trend on
26 Oct, a smaller figure with a smaller scale is added.



Furthermore, we found that the increase in SQT emission flux following irrigation
(by 545% on average) was more significant than that of the MTs (by 150% on average).
This suggests that the response of SQT emissions to water availability is stronger than that
of MTs, which could be related to the chemical properties and physiological functions of
SQTs in plants. Bonn et al. (2019) found that a sharp increase in SQT emission occurs
close to the wilting point to protect the plant against oxidative damage, as also supported
by Caser et al. (2019). The latter found that drought can induce the SQT-synthesis
mechanism. The strong increase in SQT emission after irrigation in our study further
supports the notion that enhanced synthesis of SQTs occurs shortly after the release of
drought stress.
Interestingly, the high SQT emission rates found in this study are consistent with
the findings of a previous study conducted in the same area (Li et. al., 2023), which also
reported higher emission fluxes of SQTs compared to other studies. This suggests that there
may be a unique level of drought or plant characteristics that contribute to the high emission
fluxes of SQTs in this region.

*3.3 The impact of meteorological parameters on MT and SQT emission rates under*
*drought condition*
The effect of meteorological conditions on BVOC emission rate under drought conditions
is complex and depends on many factors, including vegetation type, BVOC type, and
ambient stress. In the Eastern Mediterranean region, Li et al. (2023) found that under
drought, the best proxy for BVOC emission is the instantaneous temporal change in RH;
temporal changes in T were also better correlated with BVOC mixing ratio than absolute





values of T. Here, we examined the impact of instantaneous changes in ambient air RH and
T – $\delta_{RH}$ and $\delta_T$ , respectively (see Sect. 2.4.3), as well as of ambient air T and RH on the
BVOC emission rate. Figure 6 presents a principal component analysis (PCA) for the
correlation of both $\delta_{RH}$ and $\delta_T$ with the BVOC emission rates. Before irrigation, when the
plants were under drought, on 8 Sep, 22 Sep, 19 Oct, and 26 Oct, the emission rates of both
MTs and SQTs were better correlated with $\delta_{RH}$ and $\delta_T$ (average Pearson's value (r) of
0.56 and -0.61, respectively) than with RH and T (r of -0.22 and 0.29, respectively).
Exceptional are 14 Sep and 12 Oct, also sampled under drought conditions: on 14 Sep, the
SQT emissions showed the best correlation with RH (r = 0.97); on 12 Oct, the emission
rates of BVOCs tended not to correlate with any of the tested meteorological parameters
because of a strong correlation of T and $\delta_{RH}$ (r = -0.98).

When focusing only on the days after irrigation, except for 27 Oct, the BVOC

emissions were better correlated with T (on average, r = 0.52) than with any other
parameter. Interestingly, on 27 Oct, the SQTs tended to correlate with RH (-0.58), while
the MT emission was better correlated with $\delta_T$ (0.94). The PCA results show some
similarities between the different sampled branches, in their stronger response to $\delta_{RH}$ than
to the other tested meteorological parameters and their almost complete lack of correlation
with T when under drought conditions. However, after irrigation, all BVOC emission rates
were highly responsive to T, more than to any other parameter, reflecting the well-known
Arrhenius-type increase for BVOC emission with temperature, as mentioned in Sect. 1



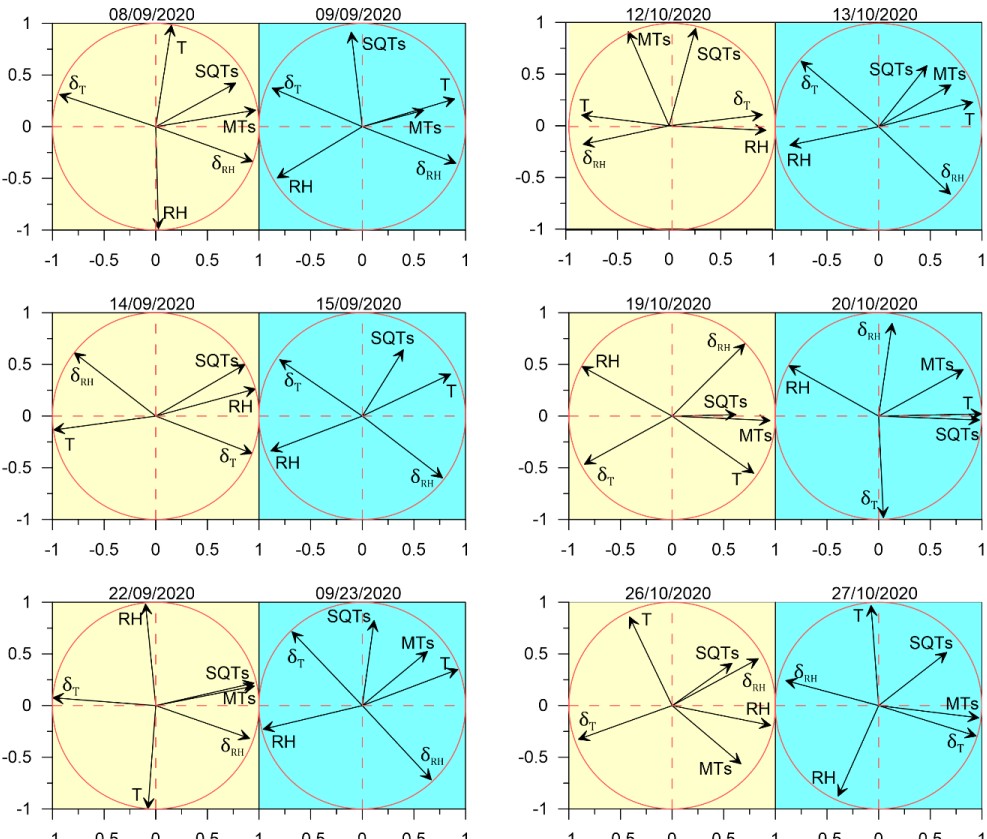

**Figure. 6** PCA analysis for the response of SQTs and MTs to meteorological parameters. The results are presented for SQTs, MTs, T, RH, $\delta_T$, and $\delta_{RH}$, individually for each measurement day. The yellow and blue shaded areas refer to the day before and after irrigation, respectively.

Table 1 summarizes the correlation coefficients between the emission rates of SQTs/MTs and RH, T, $\delta_{RH}$, and $\delta_T$, both before and after irrigation. Considering the significant variability in the emission rates of SQTs and MTs across different branches, the r values presented in the table are averages calculated from individual branch-level r values, separately before and after irrigation. Li et al. (2023) showed that under drought conditions, the temporal gradient of meteorological parameters in general was more strongly correlated



with BVOC emission rates – not only for RH, but also for T and vapor-pressure deficit.
Before irrigation, both SQT and MT emission rates were more strongly correlated with $\delta_{RH}$
and $\delta_T$ than with RH and T. However, after irrigation, the r values for the correlations with
$\delta_{RH}$ and $\delta_T$ were dramatically weakened. Moreover, following irrigation, the correlations
with T and RH for both MTs and SQTs were notably stronger than before the irrigation.
This indicates that under drought, the temporal gradients in T and RH have a stronger
impact on BVOC emissions than the absolute value of T and RH, in agreement with
findings by Li et al. (2023). Here, we demonstrate that even a relatively minor precipitation
event leads to T becoming the dominant factor in the BVOC emission rate, as expected
under non-drought conditions.  Interestingly, after irrigation, the highest r value for MTs
was with T, but for SQTs, it was with RH.

**Table 1.** Correlation between the emission rates of MTs and SQTs and the examined meteorological
parameters. Presented are the Pearson's r values for the correlation between MT/SQT emission rate and RH,
T, $\delta_{RH}$, and $\delta_T$ (green shading for SQT emissions and lavender shading for MT emissions). Blue and red
shading indicates positive and negative correlation, respectively, and the darkness of the color indicates their
values. The *P*-values for the correlation are shown in brackets.

| Pearson's r value | | | | | |
|---|---|---|---|---|---|
| **SQT** | before irrigation | after irrigation | **MT** | before irrigation | after irrigation |
| vs RH | -0.22 (0.00) | -0.46 (0.00) | vs RH | -0.18 (0.11) | -0.44 (0.04) |
| vs T | 0.33 (0.02) | 0.42 (0.00) | vs T | 0.20 (0.02) | 0.46 (0.01) |
| vs $\delta_{RH}$ | 0.53 (0.02) | -0.11 (0.00) | vs $\delta_{RH}$ | 0.54 (0.01) | 0.00 (0.00) |
| vs $\delta_T$ | -0.50 (0.02) | 0.13 (0.00) | vs $\delta_T$ | -0.48 (0.01) | 0.03 (0.00) |



The analysis presented in Fig. 6 and Table 1 reinforces the finding that
instantaneous changes in meteorological parameters, particularly $\delta_{RH}$, serve as a better
proxy for BVOC emission rate under drought conditions. This finding suggests that
modeling BVOC emission rates under drought conditions can rely on $\delta_{RH}$. In light of this
insight, we investigated the mathematical connection between $\delta_{RH}$ and the emission rates
of the MT and SQT fluxes. Exponential fitting corresponded with a relatively strong
correlation between these emission rates and $\delta_{RH}$. Other fitting types used to test this
relationship are presented in Sect. S2. Figures 7 and 8 depict the exponential fitting curves
for MTs and SQTs, respectively. These curves are presented separately for each branch
and individually for drought and post-irrigation conditions. The $r^2$ for MTs with $\delta_{RH}$ ranged
from 0.06 to 0.58 (r = 0.24–0.76, average 0.48) under drought, whereas following irrigation,
the corresponding correlations ranged from 0.02 to 0.62 (r = -0.78–0.28, average -0.08).
For SQTs, the corresponding $r^2$ values were somewhat higher, ranging from 0.04 to 0.51 (r
= -0.41–0.67, average +0.33) and 0.00 to 0.48 (r = -0.69–0.17, average -0.24), under
drought and following irrigation, respectively.
Overall, these results suggest that while $\delta_{RH}$ is likely a better proxy for MT and
SQT emission rates (see Table 1 and Sect. S3), the correlation of $\delta_{RH}$ with these BVOCs
appears to be too weak to accurately predict their emission rates using $\delta_{RH}$ values in
atmospheric modeling. Additional study is needed before $\delta_{RH}$ can effectively serve as a
parameter for modeling BVOC emission rates.

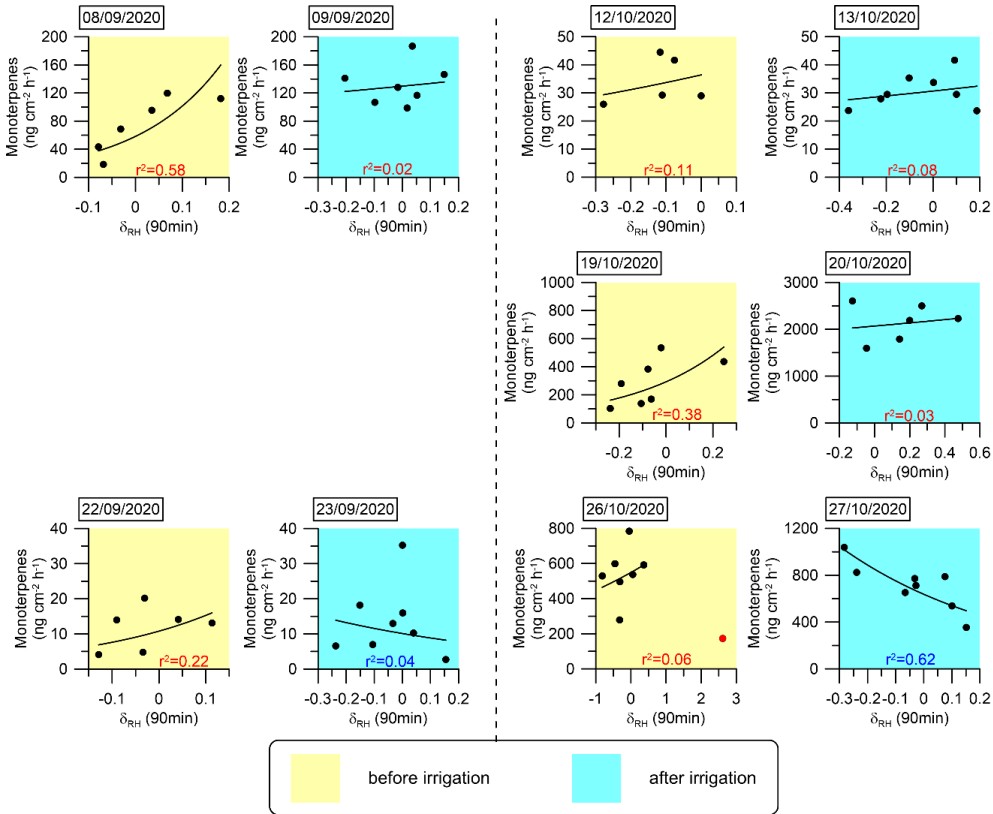

**Figure. 7** Daily correlations between MT emission fluxes and $\delta_{RH}$. An exponential fitting function was used to fit the curves. The coefficient of determination ($r^2$) for each day is marked in red or blue when the correlation is positive or negative, respectively.

Following irrigation, the correlations between the emission flux rates and $\delta_{RH}$ became more moderate (4 cases out of 11) or even negative (5 cases out of 11). This further demonstrates the high sensitivity of $\delta_{RH}$'s effect on BVOC emissions to changes in water availability. Further research is required to examine the physiological and biochemical processes underlying the sensitivity of BVOC emission rates to $\delta_{RH}$.

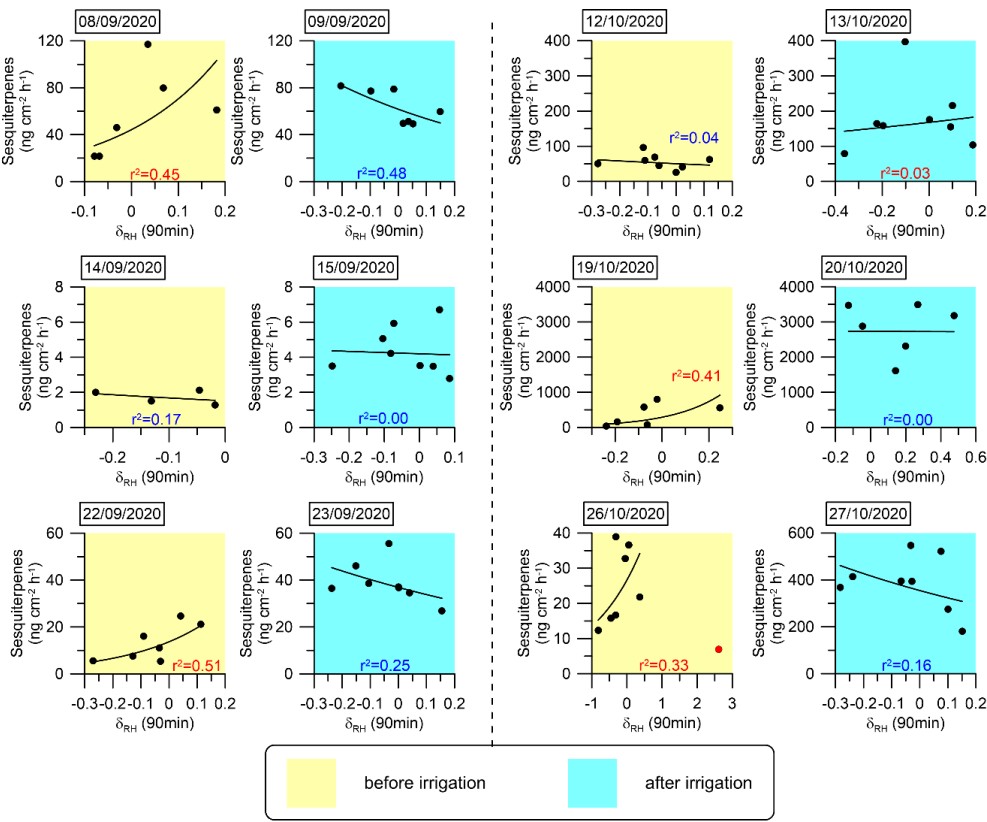

**Figure. 8** Daily correlations between SQT emission fluxes and $\delta_{RH}$. An exponential fitting function was used to fit the curves. The coefficient of determination ($r^2$) for each day is marked in red or blue when the correlation is positive or negative, respectively. The sample at 12:10 h on 26 Oct 2020 (marked in red) was not considered in the fitting curve for that day, because an extremely sharp increase in RH (from 10 to 31%) occurred within 10 min, which we considered an outlier.

## 4 Summary and conclusions

We investigated BVOC emission rates from branches of *Phillyrea latifolia* under both drought and minor irrigation conditions in the Eastern Mediterranean region, with the aim of assessing the influence of low precipitation levels and meteorological parameters on MT and SQT emission rates during drought stress. We found that leaf water content increases





gradually under prolonged periods of drought, indicating the plant's enhanced capacity for
water uptake under more severe drought conditions. The highest emission rate among all
detected MTs was of *cis*-β-ocimene, and among the detected SQTs, β-caryophyllene, α-
humulene, germacrene D, and α-farnesene. Both the MT and SQT emission rates were
significantly influenced by the availability of soil water. In response to irrigation, the MT
and SQT emission rates increased by 150% and 545%, respectively, indicating that even a
small amount of water (equivalent to 5.5–7 mm precipitation) can significantly impact their
emission rates.

This study highlights the complex way in which meteorological conditions affect

BVOC emissions under drought conditions. In line with Li et al.'s (2023) findings, under
drought, the instantaneous change of relative humidity, $\delta_{RH}$, was the best proxy for BVOC
emission rates, considering the strong correlation between MTs and SQTs and $\delta_{RH}$ (r =
0.54 and 0.53, respectively). However, after a small amount of irrigation (equivalent to
5.5–7 mm precipitation), no correlation was observed between $\delta_{RH}$ and MT emission rate,
whereas a negative correlation with $\delta_{RH}$ was observed for SQT emission rate. The increase
in soil water availability led to T (for MTs) or RH (for SQTs) becoming the dominant
meteorological parameter affecting BVOC emission rate, making them the best proxies for
BVOC emission rates among all tested meteorological parameters. This indicates that
changes in water availability can dramatically alter the manner in which BVOC emissions
respond to meteorological conditions.

Hence, according to the conditions used in this study, under more severe drought,

$\delta_{RH}$ can serve as the best proxy for BVOC emission rate, whereas under more moderate
drought, either T or RH is the best proxy for BVOCs, in agreement with previous findings



presented in the companion paper by (Li et al., 2023). Our findings indicate that even a
small amount of precipitation can lead to a transition from a drought to non-drought regime
in terms of BVOC emission rates and the manner in which they respond to meteorological
conditions.

**Author contribution**. ET designed the experiments, QL and GL carried out the field
measurements, QL performed the data acquisition. QL performed the analytical analysis
together with EB and EL. QL and ET led the data analyses with contributions from all co-
authors. QL and ET prepared the manuscript with contributions from EB.

**Competing interests.** The authors declare that they have no conflict of interest.

**Acknowledgements**
This study was supported by the Israel Science Foundation, Grant Nos. 1787/15 and
543/22. Eran Tas holds the Joseph H. and Belle R. Braun Senior Lectureship in Agriculture.

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
