# Peer review of "Impact of meteorological conditions on BVOC emission rate from Eastern Mediterranean vegetation under drought"

_EGUsphere, 2024_

## Author Comment (AC1)

We thank Referee #2 for reviewing our manuscript and for his/her critical and constructive comments. In the following, the comments by the reviewer (in italic blue font) are followed by our detailed responses.

*Li et al present experimental findings from branch enclosure measurements of BVOC emissions from Phillyrea latifolia under drought and irrigation conditions. The precise effects of meteorological conditions, under existing drought conditions, on BVOC emissions have not been well studied and this paper makes a timely contribution. The paper is clear, thorough, well-written and within the scope of Biogeosciences. I recommend publication following clarification on the below (minor) points:*

*Line 132 – 133: is there any data you can cite to support the idea that Phillyrea latifolia is the greatest BVOC-contributing plant species in the park? Or is this based on MEGAN? You mention later that the species does not emit much isoprene, so presumably this is based on MT and SQT emissions?*

**Response**: The composition of plants is unique to the specific region, and there are no other studies regarding BVOC-emission for similar shrubberies in the region. Our study focused on Phillyrea latifolia based on MEGAN results for this specific shrubbery (Li et al., 2018). The study by Li et al. (2018) also indicated negligible isoprene emission rates from this shrubbery. In our companion paper (Li et al., 2024) we reported measurement of BVOCs by Proton-transfer-reaction time-of-flight mass spectrometry (PTR-ToF-MS) in a site located 44.4 km northeast of Ramat Handiv, which comprises a similar vegetation composition .We found that in this site, isoprene has a negligible mixing ratio, in agreement with the MEGAN prediction using the specific local species composition. We have added to the revise version the study by Li et al. (2018) as a reference to the revised version.

*Line 290 – 291: the soil moisture is described as "around" and "~" but then two ranges of values to 1 decimal place are given. Could this be rephrased as "soil moisture ranged between X and Y % before irrigation, and X and Y % after irrigation".*

**Response**: Thank you for pointing out that. We have amended this in the revised version: "Soil moisture ranged between 12.5% and 14.0% before irrigation and between 14.3% and 26.2% after irrigation."

*Figure 4: On Figure 5 it's useful to have the yellow and blue shading explained in the legend, could you add that here too?*

**Response**: Thank you. This was amended as follows:

[Figure]

*Section 3.2.2 (Line 356): It is interesting to see from Figure 5 that post-irrigation, as well as an increase the amounts of SQT emitted, there are also some changes to the composition of compounds emitted – could you add some discussion around this?*

**Response**: Thank you for raising this point. It would indeed be important to analyze and discuss this finding, but we feel that we don't have enough data available for such analysis. Accordingly, we mention this finding in the revised version and state that additional study is needed to better understand the connection between irrigation during drought stress and BVOC emission composition as follows: "It is also observed that on some of the sampling days, the composition of MTs tends to become more diverse after irrigation compared to before irrigation, warranting further studies." And "In addition, the SQTs composition, like MTs composition, was observed to be more diverse after irrigation in most cases, warranting further study"

*Line 425 – 428: Is this is reason you don't present r values for the whole drought and whole irrigated sets of data that are discussed on Lines 401 – 413? If so, you could move this explanation earlier in the text to justify that.*

**Response**: Due to the large variation in BVOC emissions across different branches, the r values were calculated separately for each branch. On lines 401-413, the reported r values are calculated as the average across r values which calculated individually for each branch, for each measurement day (separately for drought and non-drought conditions) and individually for MTs and SQTs. We have added the following sentence to clarify this point: "Due to the large variation in BVOC emissions across different branches, the r values were calculated separately for each branch and each sampling day."

We have revised the sentence on lines 410-412 as follows: "the BVOC emissions were better correlated with T (averaging r values across all relevant days, r = 0.52)"

*Line 442: Please add clarification in the caption for Table 1 that these are the average Pearson coefficients from multiple individual branch values.*

**Response**: Amended: "The values are the average of r values for multiple individual branches."

References

Li Q., Gabay M., Dayan C., Misztal P., Guenther A., Fredj E., Tas E., 2024. Instantaneous intraday changes in key meteorological parameters as a proxy for the mixing ratio of BVOCs over vegetation under drought conditions. Biogeosciences.

Li Q., Gabay M., Rubin Y., Fredj E., Tas E., 2018. Measurement-based investigation of ozone deposition to vegetation under the effects of coastal and photochemical air pollution in the Eastern Mediterranean. Science of The Total Environment 645,1579–1597. https://doi.org/10.1016/j.scitotenv.2018.07.037.

---

## Author Comment (AC2)

We thank Referee #1 for reviewing our manuscript and for his/her critical and constructive comments. In the following, the comments by the reviewer (in italic blue font) are followed by our detailed responses.

*Li et al., in their article "Impact of meteorological conditions on BVOC emission rate from Eastern Mediterranean vegetation under drought", report the results from a field experiment conducted in Ramat Hanadiv Nature Park (Israel) to investigate the effect of drought on BVOC emissions from vegetation. They selected the species Phillyrea latifolia for their study. BVOC emissions from six branches were analysed between September and October 2020. After collecting BVOC emissions under natural drought conditions for a day, the plants were irrigated with a moderate amount of water, corresponding to about 5.5–7 mm of precipitation and BVOC emissions were again sampled on the following day.*

*BVOC emissions of Phillyrea latifolia were dominated by monoterpenes (MTs) and sesquiterpenes (SQTs) were also detected. The authors found that during the natural drought period, BVOC emissions were better correlated with changes in environmental parameters (especially relative humidity, RH), rather than with their absolute values.*

*The manuscript is well-structured and follows mostly a logical presentation. At times, some clarification would be needed. The results, even though derived only from one species, offer a framework that is potentially important to modelling of BVOC emissions under drought conditions. It is expected that similar experiments will be performed in the future with other species. I only have minor comments and a few technical comments (see below), which are mostly to clarify a few aspects of the method as well as to add to the discussion of the results. Therefore, I recommend accepting the manuscript with minor corrections to address the comments.*

*Minor comments:*
*l. 156: The authors write that the adsorbent tube have been 'precoated', however, I assume that the tubes are filled with adsorbent materials and not only 'coated'. Is that right?*

**Response**: Yes, thank you for pointing it out. It was a misuse of the word. We have changed the term to "filled".

*l. 185: The authors write that 'the calibration analytes were injected [...] onto clean sorbent tubes [...] at a nitrogen flow of 80 mL min$^{-1}$'. Can the authors add for how long the sorbent tubes were flushed with nitrogen after injection? And maybe the injected volume (even though it's mentioned in the supplementary material, it might be worth mentioning it here).*

**Response**: The sorbent tubes were flushed with nitrogen after injection for 5 minutes. This information as well as the injected volume have been added to the revised version.

*ll. 208-210: The way the reference samplings were performed is unclear. The authors write that 'Prior to the reference sampling, the system and branches were given at least 60 min to adapt'. Do they mean 'after the reference sample and prior to the first sampling from the branch'? Also, they write 'After the 10$^{th}$ sampling on the second measurement day [...], the*

*sampled branch was cut [...]'. Is there a reason that this was not done after the 9th sampling* *and before the last reference sample? Has the same branch been taken out of the Tedlar* *bag before the irrigation and allowed again to adapt after the first reference sampling of* *the second day? The schematic in Fig. 2 helps, but the authors could be more explicit about* *the reference samples and taking the branch in and out of the bag.*

**Response**: Thank you for the correction. There was indeed an error with the given information. On each measurement day, the adaptation time was included after the reference sampling and prior to the first branch sampling. Regarding the branch cutting, it was done after the last reference sampling. However, it is true that it could also be done between the 9th sampling, considering that the branch is not needed for the reference.
In the revised version, we specify that the branch cutting was performed after the 9th sampling on the second measurement day of each 2-sequential-day period, to avoid confusion. To address all these issues, we have revised the original text as follows:
"On each measurement day, after completing the first sampling for reference, the system and branches were allowed at least 60 minutes to adapt to the different conditions after placing the branch into the bag and setting up the equipment. At the end of the first measurement day, the sampled branch was removed from the bag and returned after the reference sampling on the second day. Following the 9th sampling on the second measurement day of each two-sequential-day period, the sampled branch was cut and sent to the laboratory for leaf analysis."

*ll. 211-216: The authors write 'Leaf net dry weight and area were evaluated within 24h', is* *that simply the net weight, as the drying occurs later, for 72h at 60°C?*

**Response**: The original sentence was incorrect. Indeed, within 24h, we only took the wet weight and scanned the leaves. The net weight was determined after drying for 72 h. The text was revised as follows:
"Leaf wet weight and area were evaluated within 24 h after cutting the branch. …. The leaves were then dried for 72 h at 60 °C, and their net dry weight was recorded."

*ll. 218-221: The authors describe which five compounds were chosen for quantification by* *GC-MS, stating that they are the ones with 'the highest sampled mass'. Could the author* *state if that is true for all individual branches or from combining all the results from all the* *branches? From Figs. 4 and 5, 'other' MTs and SQTs appear (see further comment below),* *meaning that they have also been quantified. Can the authors explain how they quantified* *the other compounds?*

**Response**: Thank you for pointing out this issue. The same five compounds were most abundant for all branches, as we specify in the revised version:
"Of the identified species, one MT and four SQT compounds (cis-β-ocimene, β-caryophyllene, α-humulene, α-farnesene, and germacrene D) with the highest sampled mass for all branches were chosen for quantification by GC–MS (see Sect. 2.3)."
In the revised manuscript, we state that "The daily average emission rate of MTs over all sampling days ranged from 11.7 to 2151.4 ng cm$^{-2}$ h$^{-1}$ (0.89–121.5 µg g$^{-1}$ h$^{-1}$), with cis-β-ocimene being the most abundant for all sampled branches, averaging at 88% of all

detected MTs". As for SQTs, we state: "The four most abundant detected SQTs for all sampled branches were β-caryophyllene, α-humulene, germacrene D, and α-farnesene. These compounds comprised 90% of all detected SQTs, from all the branches together."
The explanation about quantification for other MTs and SQTs has been added as follows: "As for the minor MTs and SQTs, the calibration curve of *cis*-β-ocimene (E, Z) and the averaged calibration curve of the four most abundant SQTs were used for a rough estimation of their emission rates."

*Sect. 2.4.3: Here I have been wondering if I understand the approach correctly. Is it so that the time of the last step (n) is the time when the BVOC sampling ends? So there are three 'steps' that are happening during the sampling, but the BVOC emission rate is an average over these 30 minutes. Have the authors considered using the time in the middle of the sampling period and have 30 minutes steps? Would these lead to similar results, but with n=3? This section might need some clarification regarding the different time steps and the assumptions made.*

**Response**: Yes, the last step (n) is when the sampling ends. In principle, if we select the time step of the meteorological parameter's measurement as 30 min instead of 10 min, with n=3, the $\delta_{RH}$ and $\delta_T$ would have similar values. However, we have chosen to calculate the $\delta_{RH}$ and $\delta_T$ using the timeframe starting 60 min prior to sampling and ending at the end of sampling because this approach was found to yield the highest correlation between $\delta_{RH}$ and $\delta_T$ and BVOCs emission rate. On lines 262-263 it is mentioned that this calculation method is based on a preliminary test. We applied 10-minute time step, rather than 30 min step, to achieve the highest accuracy for $\delta_{RH}$ and $\delta_T$. In the revised version we clarify that: we applied "10 min time step, according to the available measurement frequency….".

*l. 281 (Eq. 6): Also here is it a bit unclear what is 'n' if i=1 indicates the daily minimum. Is AET measured for each emission rate following the minimum? Should AET then by definition be always positive? Or it does not have to do with the minimum around noon-time? Or is this only valid for the drought period and for the irrigation experiment it is simply the first sample after noon? Please clarify.*

**Response**: Thanks for pointing out this confusion. AET was calculated using all measured emissions following the noontime minimum, while the minimum itself was determined as the lowest BVOC emission rate between 12:00 and 14:00.
Considering that the emission rate after 14:00 could be smaller than this minimum between 12:00 and 14:00, AET could be negative. To keep our evaluation consistent, AET for the irrigation experiment was evaluated using the same procedure. We have excluded "daily" in the phrase "the daily minimum around noontime" (lines 282-283) to avoid confusion.

*Sect. 3.3: This section would benefit from reorganization. My understanding is that PCA analysis has been done for each branch individually (Fig. 6), while the Pearson's values have been averaged? Is there a specific reason why PCA could not be done on the entire dataset? Also between lines 402 and 405, the authors mention average Person's values for MTs and SQTs with respect to δRH and δT as well as for RH and T. Are the two respective*

*values in bracket for MTs with respect to both variables and then SQTs with respect to these same two variables? This could be clarified. Also, why have the authors decided to report selected Pearson's values for the PCA and not all? They don't seem to be easily derived from Fig. 6.*

**Response**: The PCA analysis was conducted separately for each branch due to the large variation in emission rates either across branches, or due to irrigation. On some days, the emission rate ranged from 2 to 5 ng cm$^{-2}$ h$^{-1}$, while on other days, it ranged 1000-3000 ng cm$^{-2}$ h$^{-1}$. As a result, performing PCA on the entire dataset would be statistically meaningless. In accordance with the explanations given above, Pearson's values were calculated separately for each branch and each sampling day. These values were averaged across MTs SQTs, all six branches and all sampling days, resulting in 4 different values individually for each of the investigated meteorological parameters: T, RH, δRH and δT. Accordingly, we have added the following text to the revised version: "Due to the large variation in BVOC emissions across different branches, the r values were calculated separately for each branch and each sampling day."

Regarding the Pearson's values which are reported between lines 402 and 405, the r value was averaged incorporating both MTs and SQTs emission rates. In the first bracket, the two r values refer to δRH and δT, while in the second one they refer to RH and T. We have revised the sentence for clarity:

"….the emission rates of measured BVOC (including both MTs and SQTs) were better correlated with $\delta_{RH}$ and $\delta_T$ (average Pearson's value (r) of 0.56 and -0.61, respectively) than with RH and T (r of -0.22 and 0.29, respectively)."

*Fig. 6: In this figure, is it assumed that the x-axis is the first factor and the y-axis the second factor of the PCA analysis?*

**Response**: Yes, we have added this information to the axes title.

[Figure]

*ll. 462-465: How do the authors reconcile that while δRH is a better proxy (for drought conditions), the correlations are too weak to predict emissions. What about multiple regressions? Could that be an option rather than use δRH only with additional studies?*

**Response**: While our study reveals that δRH showed the highest correlation with BVOC emission rates compared with all tested meteorological parameters (as well as with vapor pressure deficit (VPD), solar radiation and their temporal gradients, which are not discussed in the manuscript), this correlation is not high enough to accurately predict the BVOCs emission rates by modelling. Additional studies may lead to improvements in utilizing our findings for modelling predictions.

VPD and its instantaneous changes showed similar but lower correlation with BVOC emission rates compare with T, RH, and their instantaneous changes. To address the reviewer's comment, we tested multiple regression by using T and RH, as well as δT and δRH, as two independent parameters. This regression was performed for each sampling day individually. However, this regression did not yield a robust parametrization, as the correlation factors varied significantly across different days. The outcome could be explained by the fact that, while δRH is the best meteorological proxy, other factors, possibly non-meteorological factors such as plant physiology, also affect BVOCs emission rates.

*ll. 496-499: Here the authors contradict themselves, calling the correlations 'strong' while previously they wrote that they were too weak to predict BVOC emissions (for drought conditions).*

Response: The terms "weak" and "strong" were used to emphasize two different arguments. "too weak to accurately predict their emission rates using $\delta_{RH}$ values in atmospheric modeling" refers to the ability of modeling to predict BVOCs emission rates based on correlation with $\delta_{RH}$, according to the level of correlation we found in our analysis. On lines 496-499, "strong correlation" refers to $\delta_{RH}$ being the best proxy for BVOC emission rates among all tested meteorological parameters, meaning that, $\delta_{RH}$ provided the best correlation with BVOCS emission rates (lines 497-498). We have rephrased the sentence on lines 497-499 to make it clearer:
"$\delta_{RH}$ was the best proxy for BVOC emission rates, as its correlation with MTs and SQTs emission rates (r = 0.54 and 0.53, respectively) was the strongest among all tested meteorological parameters."

*Technical comments:*

*l. 148: CO, HC, CO2, and H2O have not been defined previously.*

**Response**: Amended:
"oxidize carbon monoxide (CO) and hydrocarbons (HC) to carbon dioxide ($CO_2$) and water ($H_2O$)."

*Fig. 4: The legend shows 'Other monoterpene'. Is that only one other monoterpenes or various MTs? Also, the legend shows 'Medium' instead of 'Median'. Also, this figure does not include the shading in the legend, unlike Fig. 5. This could be harmonized.*

**Response**: Amended:

[Figure]

*Fig. 5: The legend in this figure also has 'Medium' instead of 'Median'.*

**Response**: Amended:

[Figure]

*ll. 640-642: I assume that this manuscript was submitted first, and the companion paper cited has been submitted later, as it seems that the title has changed and the year should be 2024 for the preprint. This should be changed in the final manuscript.*

**Response**: To be done.

*Figs. S1 to S5: It should be explained somewhere what the blue and pink arrows represent.*

**Response**: The arrows have no meaning and therefore have been removed from the figure.

*Section S2's title: There seems to be too many parentheses in this title.*

**Response**: Amended: "**S2. Linear and hyperbolic correlation between MTs/SQTs and temporal changes in RH ($\delta_{RH}$)**"

---

## Author Response (AR1)

Dear Editor,

We are pleased to submit a revised version of the manuscript (egusphere-2024-529) titled "Impact of meteorological conditions on BVOC emission rate from Eastern Mediterranean vegetation under drought" to be considered for publication in Biogeosciences. We want to thank the two reviewers for their time and efforts in reading our manuscript and for their very constructive comments. We have addressed all the reviewers' comments, which have led to improvements in both the scientific content and clarity of the manuscript. In the following, the comments by the two reviewers (in italic blue font) are followed by our detailed responses.

Sincerely,

Eran Tas

**Response to comments by reviewer #1**

*Li et al., in their article "Impact of meteorological conditions on BVOC emission rate from Eastern Mediterranean vegetation under drought", report the results from a field experiment conducted in Ramat Hanadiv Nature Park (Israel) to investigate the effect of drought on BVOC emissions from vegetation. They selected the species Phillyrea latifolia for their study. BVOC emissions from six branches were analysed between September and October 2020. After collecting BVOC emissions under natural drought conditions for a day, the plants were irrigated with a moderate amount of water, corresponding to about 5.5–7 mm of precipitation and BVOC emissions were again sampled on the following day.*

*BVOC emissions of Phillyrea latifolia were dominated by monoterpenes (MTs) and sesquiterpenes (SQTs) were also detected. The authors found that during the natural drought period, BVOC emissions were better correlated with changes in environmental parameters (especially relative humidity, RH), rather than with their absolute values.*

*The manuscript is well-structured and follows mostly a logical presentation. At times, some clarification would be needed. The results, even though derived only from one species, offer a framework that is potentially important to modelling of BVOC emissions under drought conditions. It is expected that similar experiments will be performed in the future with other species. I only have minor comments and a few technical comments (see below), which are mostly to clarify a few aspects of the method as well as to add to the discussion of the results. Therefore, I recommend accepting the manuscript with minor corrections to address the comments.*

*Minor comments:*

*l. 156: The authors write that the adsorbent tube have been 'precoated', however, I assume that the tubes are filled with adsorbent materials and not only 'coated'. Is that right?*

**Response**: Yes, thank you for pointing it out. It was a misuse of the word. We have changed the term to "filled".

*l. 185: The authors write that 'the calibration analytes were injected [...] onto clean sorbent tubes [...] at a nitrogen flow of 80 mL min$^{-1}$'. Can the authors add for how long the sorbent tubes were flushed with nitrogen after injection? And maybe the injected volume (even though it's mentioned in the supplementary material, it might be worth mentioning it here).*

**Response**: The sorbent tubes were flushed with nitrogen after injection for 5 minutes. This information as well as the injected volume have been added to the revised version. Please see line 190.

*ll. 208-210: The way the reference samplings were performed is unclear. The authors write that 'Prior to the reference sampling, the system and branches were given at least 60 min to adapt'. Do they mean 'after the reference sample and prior to the first sampling from the branch'? Also, they write 'After the 10$^{th}$ sampling on the second measurement day [...], the sampled branch was cut [...]'. Is there a reason that this was not done after the 9$^{th}$ sampling and before the last reference sample? Has the same branch been taken out of the Tedlar bag before the irrigation and allowed again to adapt after the first reference sampling of the second day? The schematic in Fig. 2 helps, but the authors could be more explicit about the reference samples and taking the branch in and out of the bag.*

**Response**: Thank you for the correction. There was indeed an error with the given information. On each measurement day, the adaptation time was included after the reference sampling and prior to the first branch sampling. Regarding the branch cutting, it was done after the last reference sampling. However, it is true that it could also be done between the 9$^{th}$ and the 10$^{th}$ sampling, considering that the branch is not needed for the reference.

In the revised version, we specify that the branch cutting was performed after the 9$^{th}$ sampling on the second measurement day of each 2-sequential-day period, to avoid confusion. The same branch has been sampled during each two sequential days. To address all these issues, we have revised the original text on lines 217-223 as follows: "On each measurement day, after completing the first sampling for reference, the system and branches were allowed at least 60 minutes to adapt to the different conditions after placing the branch into the bag and setting up the equipment. At the end of the first measurement day, the sampled branch was removed from the bag and returned after the reference sampling on the second day. Following the 9th sampling on the second measurement day of each two-sequential-day period, the sampled branch was cut and sent to the laboratory for leaf analysis."

*ll. 211-216: The authors write 'Leaf net dry weight and area were evaluated within 24h', is that simply the net weight, as the drying occurs later, for 72h at 60°C?*

**Response**: The original sentence was incorrect. Indeed, within 24h, we only measured the wet weight and scanned the leaves. The net weight was determined after drying for 72 h. The text was revised on lines 223-227 as follows:

"Leaf wet weight and area were evaluated within 24 h after cutting the branch. …. The leaves were then dried for 72 h at 60 °C, and their net dry weight was recorded."

*ll. 218-221: The authors describe which five compounds were chosen for quantification by GC-MS, stating that they are the ones with 'the highest sampled mass'. Could the author state if that is true for all individual branches or from combining all the results from all the branches? From Figs. 4 and 5, 'other' MTs and SQTs appear (see further comment below), meaning that they have also been quantified. Can the authors explain how they quantified the other compounds?*

**Response**: Thank you for pointing out this issue. The same five compounds were most abundant for each of the branches, as we specify on lines 229-232 in the revised version: "Of the identified species, one MT and four SQT compounds (cis-β-ocimene, β-caryophyllene, α-humulene, α-farnesene, and germacrene D) with the highest sampled mass for each of the branches were chosen for quantification by GC–MS (see Sect. 2.3)."

In the revised manuscript, we state on lines 322-324 that "The daily average emission rate of MTs over all sampling days ranged from 11.7 to 2151.4 ng cm$^{-2}$ h$^{-1}$ (0.89–121.5 μg g$^{-1}$ h$^{-1}$), with cis-β-ocimene being the most abundant for all sampled branches, averaging at 88% of all detected MTs". As for SQTs, we state on lines 369-371: "The four most abundant detected SQTs for each of the sampled branches were β-caryophyllene, α-humulene, germacrene D, and α-farnesene. These compounds comprised 90% of all detected SQTs, from all the branches together."

The explanation about quantification for other MTs and SQTs has been added on lines 199-202 as follows: "For the minor MTs and SQTs, the calibration curve of *cis*-β-ocimene (E, Z) and the averaged calibration curve of the four most abundant SQTs were used for a rough estimation of their emission rates."

*Sect. 2.4.3: Here I have been wondering if I understand the approach correctly. Is it so that the time of the last step (n) is the time when the BVOC sampling ends? So there are three 'steps' that are happening during the sampling, but the BVOC emission rate is an average over these 30 minutes. Have the authors considered using the time in the middle of the sampling period and have 30 minutes steps? Would these lead to similar results, but with n=3? This section might need some clarification regarding the different time steps and the assumptions made.*

**Response**: Yes, the last step (n) is when the sampling ends. In principle, if we select the time step of the meteorological parameter's measurement as 30 min instead of 10 minutes, with n=3, the $\delta_{RH}$ and $\delta_T$ would have similar values. However, we applied a 10-minute time step, rather than a 30-minute time step to achieve the highest accuracy for $\delta_{RH}$ and $\delta_T$.

On line 269 in the revised version, we clarify that: we applied "10 min time step according to the available measurement frequency….".

We have chosen to calculate the $\delta_{RH}$ and $\delta_T$ using the timeframe starting 60 minutes prior to sampling and ending at the end of sampling because this approach was found to yield the highest correlation for $\delta_{RH}$ and $\delta_T$ with BVOCs emission rate. On lines 274-275 it is mentioned that this calculation method is based on a preliminary test.

*l. 281 (Eq. 6): Also here is it a bit unclear what is 'n' if i=1 indicates the daily minimum. Is AET measured for each emission rate following the minimum? Should AET then by definition be always positive? Or it does not have to do with the minimum around noon-time? Or is this only valid for the drought period and for the irrigation experiment it is simply the first sample after noon? Please clarify.*

**Response**: Thank you for pointing out this confusion. AET was calculated using all measured emissions following the noontime minimum, while the minimum itself was determined as the lowest BVOC emission rate between 12:00 and 14:00. Considering that the emission rate after 14:00 could be smaller than this minimum between 12:00 and 14:00, AET could be negative. To keep our evaluation consistent, AET for the irrigation experiment was evaluated using the same procedure. To avoid the confusion, we have excluded "daily" in the original phrase: "…where $E_i$ is the emission rate of the $i_{th}$ sample, while $i = 1$ indicates the minimum value around noontime, between 12:00–14:00 h." (lines 294-295).

*Sect. 3.3: This section would benefit from reorganization. My understanding is that PCA analysis has been done for each branch individually (Fig. 6), while the Pearson's values have been averaged? Is there a specific reason why PCA could not be done on the entire dataset? Also between lines 402 and 405, the authors mention average Person's values for MTs and SQTs with respect to δRH and δT as well as for RH and T. Are the two respective values in bracket for MTs with respect to both variables and then SQTs with respect to these same two variables? This could be clarified. Also, why have the authors decided to report selected Pearson's values for the PCA and not all? They don't seem to be easily derived from Fig. 6.*

**Response**: The PCA analysis was conducted separately for each branch due to the large variation in emission rates either across branches, or due to irrigation. The PCA analysis was not performed on the entire dataset because the emission rate varied significantly across different days: on some days the emission rate ranged from 2 to 5 ng cm$^{-2}$ h$^{-1}$, while on other days, it ranged from 1000 to 3000 ng cm$^{-2}$ h$^{-1}$. As a result, performing PCA on the entire dataset would be statistically meaningless. In accordance with the explanations given above, Pearson's values were calculated separately for each branch and each sampling day. These values were averaged across MTs SQTs, for all six branches and all sampling days, resulting in four different values individually for each of the investigated meteorological parameters: T, RH, δRH and δT. Accordingly, we have added the following text to the revised version on lines 416-417: "Due to the large variation in BVOC emissions across

different branches, the r values were calculated separately for each branch and each sampling day."

Regarding the Pearson's values reported between lines 419 and 422 (402 and 405 in the original version), the r values were averaged incorporating both MTs and SQTs emission rates. In the first bracket, the two r values refer to δRH and δT, while in the second one they refer to RH and T. We have revised the sentence on lines 419-422 for clarity: "….the emission rates of the measured BVOC (including both MTs and SQTs) were better correlated with $\delta_{RH}$ and $\delta_T$ (average Pearson's value (r) of 0.56 and -0.61, respectively) than with RH and T (r of -0.22 and 0.29, respectively)."

We did not provide all Pearson's values for the PCA because our discussion focuses on the insights from selected branches, before and after the irrigation, highlighted by the observed correlations in the PCA. We believe that presenting all Pearson's values would be meaningless. However, in response to this comment, we are providing a table with all Pearson's values for the PCA here.

**08/09/2020**

|       | MTs   | SQTs  | T     | RH    | dT/dt | dRH/dt | VPD |
|-------|-------|-------|-------|-------|-------|--------|-----|
| MTs   |       |       |       |       |       |        |     |
| SQTs  | 0.76  |       |       |       |       |        |     |
| T     | 0.32  | 0.47  |       |       |       |        |     |
| RH    | -0.14 | -0.33 | -0.98 |       |       |        |     |
| dT/dt | -0.85 | -0.51 | 0.14  | -0.30 |       |        |     |
| dRH/dt| 0.85  | 0.54  | -0.18 | 0.34  | -0.99 |        |     |
| VPD   | 0.19  | 0.38  | 0.99  | -1.00 | 0.25  | -0.29  |     |

**12/10/2020**

|       | MTs   | SQTs  | T     | RH    | dT/dt | dRH/dt | VPD |
|-------|-------|-------|-------|-------|-------|--------|-----|
| MTs   |       |       |       |       |       |        |     |
| SQTs  | 0.81  |       |       |       |       |        |     |
| T     | 0.55  | -0.21 |       |       |       |        |     |
| RH    | -0.50 | 0.25  | -0.98 |       |       |        |     |
| dT/dt | -0.25 | 0.23  | -0.62 | 0.75  |       |        |     |
| dRH/dt| 0.29  | -0.19 | 0.60  | -0.73 | -0.99 |        |     |
| VPD   | 0.49  | -0.27 | 0.99  | -1.00 | -0.68 | 0.66   |     |

**09/09/2020**

|       | MTs   | SQTs  | T     | RH    | dT/dt | dRH/dt | VPD |
|-------|-------|-------|-------|-------|-------|--------|-----|
| MTs   |       |       |       |       |       |        |     |
| SQTs  | 0.06  |       |       |       |       |        |     |
| T     | 0.42  | 0.04  |       |       |       |        |     |
| RH    | -0.41 | -0.30 | -0.96 |       |       |        |     |
| dT/dt | -0.40 | 0.32  | -0.60 | 0.50  |       |        |     |
| dRH/dt| 0.42  | -0.33 | 0.67  | -0.54 | -0.99 |        |     |
| VPD   | 0.41  | 0.20  | 0.99  | -0.99 | -0.52 | 0.58   |     |

**13/10/2020**

|       | MTs   | SQTs  | T     | RH    | dT/dt | dRH/dt | VPD |
|-------|-------|-------|-------|-------|-------|--------|-----|
| MTs   |       |       |       |       |       |        |     |
| SQTs  | 0.48  |       |       |       |       |        |     |
| T     | 0.65  | 0.39  |       |       |       |        |     |
| RH    | -0.49 | -0.33 | -0.97 |       |       |        |     |
| dT/dt | -0.25 | -0.07 | -0.49 | 0.50  |       |        |     |
| dRH/dt| 0.30  | 0.07  | 0.39  | -0.35 | -0.95 |        |     |
| VPD   | 0.54  | 0.33  | 0.99  | -0.99 | -0.47 | 0.34   |     |

**14/09/2020**

|       | MTs | SQTs  | T     | RH    | dT/dt | dRH/dt | VPD |
|-------|-----|-------|-------|-------|-------|--------|-----|
| MTs   |     |       |       |       |       |        |     |
| SQTs  | -   |       |       |       |       |        |     |
| T     | -   | -0.92 |       |       |       |        |     |
| RH    | -   | 0.97  | -0.98 |       |       |        |     |
| dT/dt | -   | 0.62  | -0.88 | 0.80  |       |        |     |
| dRH/dt| -   | -0.38 | 0.69  | -0.60 | -0.95 |        |     |
| VPD   | -   | -0.95 | 0.99  | -1.00 | -0.83 | 0.63   |     |

**19/10/2020**

|       | MTs   | SQTs  | T     | RH    | dT/dt | dRH/dt | VPD |
|-------|-------|-------|-------|-------|-------|--------|-----|
| MTs   |       |       |       |       |       |        |     |
| SQTs  | 0.97  |       |       |       |       |        |     |
| T     | 0.74  | 0.66  |       |       |       |        |     |
| RH    | -0.87 | -0.79 | -0.96 |       |       |        |     |
| dT/dt | -0.77 | -0.64 | -0.47 | 0.54  |       |        |     |
| dRH/dt| 0.62  | 0.58  | 0.21  | -0.27 | -0.92 |        |     |
| VPD   | 0.84  | 0.76  | 0.97  | -0.99 | -0.47 | 0.21   |     |

**15/09/2020**

|       | MTs | SQTs  | T     | RH    | dT/dt | dRH/dt | VPD |
|-------|-----|-------|-------|-------|-------|--------|-----|
| MTs   |     |       |       |       |       |        |     |
| SQTs  | -   |       |       |       |       |        |     |
| T     | -   | 0.40  |       |       |       |        |     |
| RH    | -   | -0.37 | -0.97 |       |       |        |     |
| dT/dt | -   | -0.10 | -0.39 | 0.47  |       |        |     |
| dRH/dt| -   | 0.01  | 0.39  | -0.43 | -0.95 |        |     |
| VPD   | -   | 0.37  | 0.99  | -1.00 | -0.42 | 0.39   |     |

**20/10/2020**

|       | MTs   | SQTs  | T     | RH    | dT/dt | dRH/dt | VPD |
|-------|-------|-------|-------|-------|-------|--------|-----|
| MTs   |       |       |       |       |       |        |     |
| SQTs  | 0.65  |       |       |       |       |        |     |
| T     | 0.67  | 0.96  |       |       |       |        |     |
| RH    | -0.41 | -0.89 | -0.84 |       |       |        |     |
| dT/dt | -0.16 | -0.17 | -0.29 | -0.27 |       |        |     |
| dRH/dt| 0.13  | 0.33  | 0.35  | 0.12  | -0.83 |        |     |
| VPD   | 0.50  | 0.93  | 0.90  | -0.99 | 0.14  | -0.02  |     |

**22/09/2020**

|       | MTs   | SQTs  | T     | RH    | dT/dt | dRH/dt | VPD |
|-------|-------|-------|-------|-------|-------|--------|-----|
| MTs   |       |       |       |       |       |        |     |
| SQTs  | 0.15  |       |       |       |       |        |     |
| T     | 0.03  | 0.63  |       |       |       |        |     |
| RH    | -0.05 | -0.60 | -0.99 |       |       |        |     |
| dT/dt | -0.52 | -0.75 | -0.83 | 0.78  |       |        |     |
| dRH/dt| 0.37  | 0.73  | 0.80  | -0.73 | -0.99 |        |     |
| VPD   | 0.02  | 0.62  | 0.99  | -1.00 | -0.79 | 0.75   |     |

**26/10/2020**

|       | MTs   | SQTs  | T     | RH    | dT/dt | dRH/dt | VPD |
|-------|-------|-------|-------|-------|-------|--------|-----|
| MTs   |       |       |       |       |       |        |     |
| SQTs  | 0.29  |       |       |       |       |        |     |
| T     | -0.63 | 0.09  |       |       |       |        |     |
| RH    | 0.67  | 0.37  | -0.57 |       |       |        |     |
| dT/dt | -0.32 | -0.52 | 0.13  | -0.84 |       |        |     |
| dRH/dt| 0.28  | 0.48  | 0.06  | 0.73  | -0.94 |        |     |
| VPD   | -0.75 | -0.23 | 0.81  | -0.94 | 0.65  | -0.51  |     |

**23/09/2020**

|       | MTs   | SQTs  | T     | RH    | dT/dt | dRH/dt | VPD |
|-------|-------|-------|-------|-------|-------|--------|-----|
| MTs   |       |       |       |       |       |        |     |
| SQTs  | 0.23  |       |       |       |       |        |     |
| T     | 0.73  | 0.39  |       |       |       |        |     |
| RH    | -0.65 | -0.32 | -0.98 |       |       |        |     |
| dT/dt | 0.00  | 0.43  | -0.37 | 0.49  |       |        |     |
| dRH/dt| -0.01 | -0.42 | 0.36  | -0.47 | -0.99 |        |     |
| VPD   | 0.71  | 0.36  | 1.00  | -0.99 | -0.42 | 0.40   |     |

**27/10/2020**

|       | MTs   | SQTs  | T     | RH    | dT/dt | dRH/dt | VPD |
|-------|-------|-------|-------|-------|-------|--------|-----|
| MTs   |       |       |       |       |       |        |     |
| SQTs  | 0.64  |       |       |       |       |        |     |
| T     | -0.19 | 0.35  |       |       |       |        |     |
| RH    | -0.22 | -0.58 | -0.83 |       |       |        |     |
| dT/dt | 0.94  | 0.40  | -0.32 | -0.13 |       |        |     |
| dRH/dt| -0.83 | -0.30 | 0.26  | 0.23  | -0.94 |        |     |
| VPD   | 0.09  | 0.51  | 0.93  | -0.98 | -0.01 | -0.08  |     |

*Fig. 6: In this figure, is it assumed that the x-axis is the first factor and the y-axis the second factor of the PCA analysis?*

**Response**: Yes, we have added this information to the axes title.

[Figure]

*ll. 462-465: How do the authors reconcile that while δRH is a better proxy (for drought conditions), the correlations are too weak to predict emissions. What about multiple regressions? Could that be an option rather than use δRH only with additional studies?*

**Response**: While our study reveals that δRH showed the highest correlation with BVOC emission rates compared with all tested meteorological parameters (in addition to vapor pressure deficit (VPD), solar radiation and their temporal gradients, which are not discussed in the manuscript), this correlation is not high enough to accurately predict BVOCs emission rates by modelling. Additional studies may lead to improvements in utilizing our findings for modelling predictions.

VPD and its instantaneous changes showed a similar but lower correlation with BVOC emission rates compare to T, RH, and their instantaneous changes. To address the reviewer's comment, we tested multiple regression by using two sets: T with RH, as well as δT with δRH, each as two independent parameters. These regressions were performed for each sampling day individually. However, these regressions did not yield a robust

parametrization, as the correlation coefficients varied significantly across different days. This outcome could be explained by the fact that, while δRH is the best meteorological proxy, other factors, possibly non-meteorological factors such as plant physiology, also affect BVOCs emission rates.

*ll. 496-499: Here the authors contradict themselves, calling the correlations 'strong' while previously they wrote that they were too weak to predict BVOC emissions (for drought conditions).*

**Response**: The terms "weak" and "strong" were used to emphasize two different arguments. "too weak to accurately predict their emission rates using $\delta_{RH}$ values in atmospheric modeling" refers to the modeling's ability to predict BVOCs emission rates based on correlation with $\delta_{RH}$, according to the level of correlation we found in our analysis. On lines 515-517 (lines 496-499 in the original version), "strong correlation" refers to $\delta_{RH}$ being the best proxy for BVOC emission rates among all tested meteorological parameters, indicating that $\delta_{RH}$ provided the strongest correlation with BVOCS emission rates (lines 515-517). We have rephrased the sentence on lines 515-517 to make it clearer:
"…$\delta_{RH}$ was the best proxy for BVOC emission rates, as its correlation with MTs and SQTs emission rates (r = 0.54 and 0.53, respectively) was the strongest among all tested meteorological parameters."

*Technical comments:*

*l. 148: CO, HC, CO2, and H2O have not been defined previously.*

**Response**: Amended on lines 152-153:
"…oxidize carbon monoxide (CO) and hydrocarbons (HC) to carbon dioxide ($CO_2$) and water ($H_2O$)."

*Fig. 4: The legend shows 'Other monoterpene'. Is that only one other monoterpenes or various MTs? Also, the legend shows 'Medium' instead of 'Median'. Also, this figure does not include the shading in the legend, unlike Fig. 5. This could be harmonized.*

**Response**: Amended:

[Figure]

*Fig. 5: The legend in this figure also has 'Medium' instead of 'Median'.*

**Response**: Amended:

[Figure]

*ll. 640-642: I assume that this manuscript was submitted first, and the companion paper cited has been submitted later, as it seems that the title has changed and the year should be 2024 for the preprint. This should be changed in the final manuscript.*

**Response**: Amended.

*Figs. S1 to S5: It should be explained somewhere what the blue and pink arrows represent.*

**Response**: The arrows have no meaning and therefore have been removed from the figure.

*Section S2's title: There seems to be too many parentheses in this title.*

**Response**: Amended: "**S2. Linear and hyperbolic correlation between MTs/SQTs and temporal changes in RH ($\delta_{RH}$)**"

**Response to comments by reviewer #2**

*Li et al present experimental findings from branch enclosure measurements of BVOC emissions from Phillyrea latifolia under drought and irrigation conditions. The precise effects of meteorological conditions, under existing drought conditions, on BVOC emissions have not been well studied and this paper makes a timely contribution. The paper is clear, thorough, well-written and within the scope of Biogeosciences. I recommend publication following clarification on the below (minor) points:*

*Line 132 – 133: is there any data you can cite to support the idea that Phillyrea latifolia is the greatest BVOC-contributing plant species in the park? Or is this based on MEGAN? You mention later that the species does not emit much isoprene, so presumably this is based on MT and SQT emissions?*

**Response**: The composition of plants is unique to the specific measurement site, and there are no studies regarding BVOC-emissions for similar shrubberies in the region, except for the MEGAN simulations for this site (Ramat Hanadiv) by Li et al. (2018). These MEGAN simulations indicated that Phillyrea latifolia accounts for the highest overall BVOC emissions, including, indeed, MTs and SQTs (Li et al., 2018). Therefore, we focused our study on Phillyrea latifolia. The study by Li et al. (2018) also indicated negligible isoprene emission rates from this shrubbery, based on MEGAN simulations. In our companion paper (Li et al., 2024), we reported measurement of BVOCs by Proton-transfer-reaction time-of-flight mass spectrometry (PTR-ToF-MS) at a site located 44.4 km northeast of Ramat Handiv, which comprises a similar vegetation composition.We found that in this site, isoprene has a negligible mixing ratio, in agreement with the MEGAN prediction using the specific local species composition. We have added the study by Li et al. (2018) as a reference in section 2.1 of the revised version.

*Line 290 – 291: the soil moisture is described as "around" and "~" but then two ranges of values to 1 decimal place are given. Could this be rephrased as "soil moisture ranged between X and Y % before irrigation, and X and Y % after irrigation".*

**Response**: Thank you for pointing that out. We have amended this on lines 301-302 in the revised version: "Soil moisture ranged between 12.5% and 14.0% before irrigation and between 14.3% and 26.2% after irrigation."

*Figure 4: On Figure 5 it's useful to have the yellow and blue shading explained in the legend, could you add that here too?*

**Response**: Thank you. This was amended as follows:

[Figure]

*Section 3.2.2 (Line 356): It is interesting to see from Figure 5 that post-irrigation, as well as an increase the amounts of SQT emitted, there are also some changes to the composition of compounds emitted – could you add some discussion around this?*

**Response**: Thank you for raising this point. It would indeed be important to analyze and discuss this finding, but we feel that we do not have enough data available for such analysis. Accordingly, we mention this finding in the revised version and state that additional study is needed to better understand the connection between irrigation during drought stress and BVOC emission composition as follows (line 348-350): "It is also observed that on some of the sampling days, the composition of MTs tends to become more diverse after irrigation compared to before irrigation, warranting further studies." And on line 392-393: "In addition, the SQTs composition, like MTs composition, was observed to be more diverse after irrigation in most cases, warranting further study."

*Line 425 – 428: Is this is reason you don't present r values for the whole drought and whole irrigated sets of data that are discussed on Lines 401 – 413? If so, you could move this explanation earlier in the text to justify that.*

**Response**: Due to the large variation in BVOC emissions across different branches, the r values were calculated separately for each branch. On lines 418-426 (lines 401-413 in the original version), we reported the r values as the average of the r values calculated individually for each branch, for each measurement day (separately for drought and non-drought conditions) and individually for MTs and SQTs. We have added the following

sentence on lines 416-417 to clarify this point: "Due to the large variation in BVOC emissions across different branches, the r values were calculated separately for each branch and each sampling day."

To make this point clearer, we have revised the sentence on lines 427-429 as follows: "…the BVOC emissions were better correlated with T (averaging r values across all relevant days, r = 0.52) than with any other parameter."

*Line 442: Please add clarification in the caption for Table 1 that these are the average Pearson coefficients from multiple individual branch values.*

**Response**: Amended: "The values are the average of r values across multiple individual branches."

**References**

Li Q., Gabay M., Dayan C., Misztal P., Guenther A., Fredj E., Tas E., 2024. Instantaneous intraday changes in key meteorological parameters as a proxy for the mixing ratio of BVOCs over vegetation under drought conditions. Biogeosciences.

Li Q., Gabay M., Rubin Y., Fredj E., Tas E., 2018. Measurement-based investigation of ozone deposition to vegetation under the effects of coastal and photochemical air pollution in the Eastern Mediterranean. Science of The Total Environment 645,1579–1597. https://doi.org/10.1016/j.scitotenv.2018.07.037.